# Next-Generation Sequencing for HIV Drug Resistance Testing: Laboratory, Clinical, and Implementation Considerations

**DOI:** 10.3390/v12060617

**Published:** 2020-06-05

**Authors:** Santiago Ávila-Ríos, Neil Parkin, Ronald Swanstrom, Roger Paredes, Robert Shafer, Hezhao Ji, Rami Kantor

**Affiliations:** 1Centre for Research in Infectious Diseases, National Institute of Respiratory Diseases, Calzada de Tlalpan 4502, Col. Sección XVI, Mexico City 14080, Mexico; 2Data First Consulting, Inc., Sebastopol, CA 95472, USA; nparkin34@gmail.com; 3Lineberger Comprehensive Cancer Center, University of North Carolina at Chapel Hill, Chapel Hill, NC 27599, USA; ron_swanstrom@med.unc.edu; 4IrsiCaixa AIDS Research Institute, Badalona, 08916 Catalonia, Spain; rparedes@irsicaixa.es; 5Division of Infectious Diseases, Stanford University School of Medicine, Stanford, CA 94305, USA; rshafer@stanford.edu; 6National HIV and Retrovirology Laboratories at JC Wilt Infectious Diseases Research Centre, Public Health Agency of Canada, Winnipeg, MB R3E 3R2, Canada; hezhao.ji@canada.ca; 7Department of Medical Microbiology and Infectious Diseases, University of Manitoba, Winnipeg, MB R3E 0J9, Canada; 8Division of Infectious Diseases, Brown University Alpert Medical School, Providence, RI 02906, USA; rkantor@brown.edu

**Keywords:** HIV drug resistance, next-generation sequencing, low/medium-income countries, implementation, low-abundance variants

## Abstract

Higher accessibility and decreasing costs of next generation sequencing (NGS), availability of commercial kits, and development of dedicated analysis pipelines, have allowed an increasing number of laboratories to adopt this technology for HIV drug resistance (HIVDR) genotyping. Conventional HIVDR genotyping is traditionally carried out using population-based Sanger sequencing, which has a limited capacity for reliable detection of variants present at intra-host frequencies below a threshold of approximately 20%. NGS has the potential to improve sensitivity and quantitatively identify low-abundance variants, improving efficiency and lowering costs. However, some challenges exist for the standardization and quality assurance of NGS-based HIVDR genotyping. In this paper, we highlight considerations of these challenges as related to laboratory, clinical, and implementation of NGS for HIV drug resistance testing. Several sources of variation and bias occur in each step of the general NGS workflow, i.e., starting material, sample type, PCR amplification, library preparation method, instrument and sequencing chemistry-inherent errors, and data analysis options and limitations. Additionally, adoption of NGS-based HIVDR genotyping, especially for clinical care, poses pressing challenges, especially for resource-poor settings, including infrastructure and equipment requirements and cost, logistic and supply chains, instrument service availability, personnel training, validated laboratory protocols, and standardized analysis outputs. The establishment of external quality assessment programs may help to address some of these challenges and is needed to proceed with NGS-based HIVDR genotyping adoption.

## 1. Introduction

Conventional HIV drug resistance (HIVDR) genotyping, in the context of clinical care, research and public health, is traditionally carried out using population-based Sanger sequencing techniques, which have a limited capacity for reliable detection of variants present at intra-host frequencies below a threshold of approximately 20% [1,2,3,4]. Several studies have suggested that low-abundance HIVDR variants could have a relevant clinical impact and that their detection could improve treatment outcomes [5,6,7,8,9]. However, the clinically relevant threshold is still to be defined and may be drug class, drug, or even mutation-dependent. With the widespread use and decreasing costs of next generation sequencing (NGS) techniques, availability of commercial kits, and development of dedicated and freely available analysis pipelines [4,10], an increasing number of laboratories around the world, especially in high-income contexts are considering or definitively moving toward these technologies for HIVDR testing [11]. NGS enables high-throughput, massively parallel sequencing of individual input templates, with the potential to improve sensitivity and quantitatively identify low-abundance variants. However, many challenges exist for the generalized adoption of NGS for HIVDR testing for clinical care, research and public health purposes, including infrastructure requirements, reagents/service providers/distributors, availability of trained personnel, fully validated laboratory protocols, standardized analysis outputs, and external quality assessment (EQA) programs. In this article, we review issues and considerations for the use of NGS for HIVDR testing from the laboratory, clinical, and implementation points of view. Some of the issues raised in this paper may seem obvious or not novel for experienced laboratories, especially in developed countries. However, we believe that these points need to be underscored for developing countries who are planning to implement NGS technologies locally, as some issues can be overlooked. We also believe that many of the methodological, sample-related, and analysis-associated issues raised in the manuscript are current problems being faced by experts in the field, and are also of interest to more advanced groups and laboratories.

## 2. Laboratory Considerations

NGS generates millions of small sequencing reads from the input templates in parallel, which provide both sequencing depth and increased sensitivity to detect low-abundance variants [1,11]. In the case of HIV, NGS allows for simultaneous sequencing of all genetic regions of interest or even the complete viral genome, representing a potential for a lower per-sample cost if multiplexing is used. Multiplexing is achieved through the introduction of unique barcoding sequences to the sequencing library of each specimen included in a sequencing run.

The market of HIVDR testing based on NGS technologies is currently dominated by second-generation [12] NGS platforms, namely Illumina and Ion Torrent. The general NGS workflow for HIVDR testing involves a set of general steps performed in a wet laboratory space, including nucleic acid extraction, PCR amplification of the HIV genes of interest, library preparation, and sequencing. The process is followed by a data analysis step that can be carried out in a dry laboratory area or office space. Each of these steps introduces variation and possible bias that affects the NGS output representativeness of the viral quasispecies, HIV drug resistance mutation (DRM) detection sensitivity and accuracy, and subsequently HIVDR interpretation, which will be discussed in the following sections (Figure 1). Several commercial options for HIVDR testing using NGS are already available and have obtained regulatory approval as in vitro diagnostic (IVD) products by reference agencies (Table 1) [13,14,15]. However, due to cost and flexibility issues, many laboratories have opted for developing and validating in house protocols. Implementation of *in house* protocols has been greatly facilitated with the accessibility to freely available public pipelines specifically designed for HIVDR analysis from NGS data [10]. Nevertheless, several challenges remain to achieve standardization in sample processing, data analysis, HIVDR reporting, and quality assurance, all of which have important implications for the adoption of NGS technologies for clinical care, research, and public health applications.

### 2.1. Sample Type and Nucleic Acid Extraction

The starting material, i.e., plasma viral RNA versus proviral DNA obtained from peripheral blood mononuclear cells (PBMC) or the buffy coat fraction, can strongly influence the detection of DRMs and HIVDR test results [16]. Population sequencing on proviral DNA can introduce bias due to the inclusion of defective virus sequences, leading to an increased proportion of hypermutation and stop codons [17,18], especially in samples where the contribution of proviral DNA outweighs that of plasma virus RNA. Such bias is particularly significant for specimens with low viral load.

Collection of dried blood spots (DBS) for HIV genotyping has become increasingly popular in low-resource settings, due to advantages in storage and shipment of DBS compared to plasma samples [19], and thanks to efforts by the World Health Organization (WHO) to provide standardization and EQA for the use of this alternative [20]. HIVDR results from DBS have been shown to be highly equivalent to those obtained from plasma [21,22,23,24], although differences do occasionally arise that can affect HIVDR interpretation [25,26]. Nucleic acids extracted from DBS samples comprise an HIV RNA component from the replication-competent plasma virus and a DNA component from cells containing proviral DNA. The contribution of proviral DNA to the sequences can become important at lower viral load values. Indeed, previous studies have reported that plasma/DBS concordance is highest when viral load is ≥5000 copies/mL, the patient has no antiretroviral therapy (ART) exposure and the duration of HIV infection is ≤2 years [27]. Furthermore, storage conditions may also significantly impact nucleic acid integrity and thus influence the representation of the type of nucleic acid being amplified, affecting the HIVDR profiles detected [19]. Special consideration should be made to samples with low HIV RNA copy number, in which genetic diversity is inherently low. The use of low thresholds to call HIVDR variants in these types of samples could lead to artefactual diversity.

### 2.2. PCR Amplification

PCR errors, including nucleotide misincorporation and PCR-mediated recombination, introduce artefactual diversity into the sequence population. By contrast, PCR resampling, i.e., repetitive amplification of the same original template, could introduce artificial homogeneity and misrepresentation of specific variants in the viral pool [28]. Recommendations on the use of primer identifier (ID) sequences have been made to account for PCR resampling and accurately define the number of input templates being sequenced, greatly reducing the error rate of NGS of HIV genomic RNA populations [28]. The Primer ID strategy involves introducing a degenerate nucleotide block in the cDNA synthesis primer, allowing each of the original template copies to have its own identifier. Reads with the same Primer ID after the sequencing step indicate PCR resampling of the same original cDNA template. The advantages and drawbacks of the use of Primer ID approaches have been thoroughly described by Swanstrom et al. in this special issue. Others have opted for quantifying input cDNA copy numbers [29]. Several additional approaches have been suggested to lower the likelihood of occurrence of PCR errors, such as the use of high fidelity PCR enzymes [30], or reduction of the number of PCR cycles [31]. Analyzing the co-occurrence of low-abundance variants in the same read could also be useful to ascertain the presence of each variant independently. However, this approach is limited to short genetic distances as the reads from the most commonly used NGS platforms are relatively short. Amplification may also be skewed by polymorphisms associated with HIVDR which may affect primer annealing in the assay [32,33], although this is a generic problem for all cDNA synthesis and PCR strategies. When possible, primer design should also consider the local viral diversity in order to optimize annealing. The length of the HIV genetic region to be amplified may also cause bias; firstly, due to the possible need of enzymes with higher processivity, but inherently lower fidelity, and secondly, because of a decrease in the number of cDNA molecules that can be generated when attempting to amplify longer genetic sections.

### 2.3. Library Preparation

The method used for library preparation for NGS may also influence variant representation in the sample. Library building methods are mostly based on tagmentation and enzymatic fragmentation of the target DNA, with multiple commercial options available (Table 2). Tagmentation is a common method involving simultaneous DNA fragmentation and insertion of short double-stranded oligonucleotide sequences to the ends of the cleaved DNA by a transposase in a single step [34]. This method is fast and has low input DNA requirements, but involves a subsequent enrichment PCR amplification step with a small number of amplification cycles to introduce sequencing adapters. Alternatively, enzymatic fragmentation methods with subsequent end repair, dA-tailing, and adapter ligation can be used. These methods generally require higher DNA inputs considering sample loss because of clean-up steps between enzymatic reactions. Nevertheless, optimized methodologies have been developed to perform all the necessary enzymatic reactions in the same vial and avoiding mechanical DNA shearing workflows, significantly lowering sample loss and DNA damage with a consequent lower input DNA requirement [35]. Less frequently, capture methods can be used, designing a set of labeled oligonucleotide probes that can bind to the genetic regions of interest in order to selectively sequester them for further sequencing, skipping the initial PCR amplification step. However, these methods require longer hands-on time, are more expensive, and are less suitable for high-throughput work.

### 2.4. Sequencing

Although multiple NGS instruments have been used for HIVDR genotyping, according to laboratory capacity, test demand, and harmonization with other NGS applications, two instruments dominate the current market for this application: the Illumina MiSeq and the Ion Torrent Personal Genome Machine (Table 3).

Sequencing errors can also contribute to artefactual sequence diversity and are inherent to specific sequencing chemistries and platforms [36] (Table 3). Sequencing errors are most relevant in samples for which the read coverage is too low to achieve the redundancy required to prevent these errors from being identified at a low threshold [37]; e.g., with a coverage of 1000 reads, the same random error would have to occur 10 times to be detected as a sequencing artefact at 1% threshold, but with a coverage of 100 reads, an error present just once would be detected as a sequencing artefact at 1% threshold. In general, lowering technical cut-off values below 1% could result in a significant increase in false positives due to inherent errors from the assay [37]. However, the number of actual input HIV templates analyzed could significantly affect results and limit the minimal cut-off value from which drug resistant mutations can be confidently called, due to the problem of PCR resampling cited above. This is especially relevant for samples with low viral load or where a limited number of input DNA templates is being analyzed due to suboptimal sample handling or to specific assay characteristics. It is important to keep in mind that technical thresholds to call low-abundance variants accurately are completely independent of the clinically relevant threshold of low-abundance DRM, which remains to be defined. Even if the accuracy to call low-abundance variants were high at very low thresholds, these variants may or may not be of clinical significance.

An interesting approach has been described to determine the optimal low-abundance variant detection threshold in a sample-specific manner, based on the post hoc analysis of the proportion of unusual mutations (defined as having a prevalence <0.01% in global HIV-1 group M sequences) or signature apolipoprotein B mRNA editing enzyme catalytic polypeptide-like (APOBEC)-mediated G-to-A hypermutation in the sample at different detection thresholds [37]. This approach could be useful to avoid selecting a threshold that is too low for a particular sample, posing an unacceptable risk of identifying artefactual mutations.

### 2.5. Analysis Pipelines

Several analysis pipelines have been independently developed to address the needs of automation and standardization for NGS-based HIVDR testing. A recent set of recommendations for bioinformatics pipelines was proposed by a group of experts in the First “Winnipeg Consensus” regarding some of the most common sources of analysis-associated variation and bias: NGS read quality control/quality assurance, NGS read alignment and reference mapping, HIV variant calling and variant quality control, HIVDR interpretation and reporting, and general analysis data management [10]. These recommendations could serve as an initial baseline to standardize NGS data analysis for HIVDR testing, aiming to optimize and refine existing pipelines and those in development. Recent work comparing the performance of several specific pipelines for HIVDR NGS data, namely HyDRA, MiCall, PASeq, Hivmmer, and DEEPGEN, demonstrated a dramatic fall in specificity for calling low-abundance variants under the 2% threshold [4]. These differences were most likely associated with each pipeline’s NGS read quality control strategies.

## 3. Use of NGS for Clinical HIVDR Testing

Sanger sequencing is still the mainstay for clinical HIVDR testing. However, the main drawback of this technology is its inability to reliably detect low-abundance DRMs below the 20% threshold [1,39,40,41]. Several studies have shown that NGS-based HIVDR testing is highly concordant to Sanger sequencing at a 20% detection threshold [5,15,16,42,43]. Thus, leveraging on the potentially improved efficiency, increased scalability, decreased cost, and higher sensitivity, many laboratories are transitioning to NGS for HIVDR testing. However, the use of NGS for HIVDR testing in clinical care involves a higher level of rigor, compared to its public health or research applications, to allow clinicians to act on the results for patient care. As technology becomes more cost-effective, a higher level of standardization in laboratory protocols, data processing, and reporting is necessary. Additionally, when available, steady access to and active participation in an accredited EQA program would be essential [44].

Although NGS can potentially improve the detection of DRMs present under the 20% threshold [41], the clinical relevance of this increased sensitivity is yet unknown. Moreover, the time to failure associated with different low-abundance drug resistance variants at different thresholds needs to be assessed to avoid too early or unnecessary antiretroviral drug switching, especially in the context of low- and medium-income countries (LMIC) where limited treatment options exist. Indeed, a recent meta-analysis showed that high heterogeneity of study design and methodologies complicates standardization and measurement of the clinical impact of low-abundance HIVDR variants [33]. Previous studies have suggested that a 5% threshold could provide a reproducible clinically relevant correlation with treatment outcome [5,15,45,46]. However, the clinical benefit of increased sensitivity does not yet outweigh the need for standardization of NGS.

While a clinically relevant threshold is defined and standardized methods to accurately and reproducibly call low-abundance variants are agreed upon, NGS-based HIVDR methods could be validated as equivalent to the Sanger sequencing gold standard at the 20% threshold. Theoretically, this would provide increased efficiency and lower cost, without the increased sensitivity advantage of NGS. Of note, the trade-off between cost and processing speed could significantly hinder NGS adoption for clinical HIVDR testing, as larger batch sizes need to be gathered in order to decrease sequencing cost, at the expense of turnaround time. NGS samples will take twice as long to pass through the process, which can be even more significant if repeat testing is necessary.

Moreover, reports from NGS data could initially simulate Sanger sequencing output for easier adoption and interpretation by clinicians [10]. This is especially important in LMIC, where doctors are often not trained to interpret and act upon HIV genotyping data, and even less if the complexity of low-abundance variant reporting is added.

When looking for answers to standardize and implement NGS-based HIVDR testing for clinical purposes, experiences in other fields could be highly valuable [38,47]. Reaching consensus on standardized reporting formats with minimal requirements, terminology, and variables could provide a working frame to achieve standardization. Additionally, an effective EQA program could help in addressing clinical challenges, with the use of both wet and dry proficiency evaluation panels. This topic is thoroughly evaluated by Lee et al., Noguera-Julian et al., and Ji et al., as part of this special issue [44,48,49].

## 4. NGS-Based HIVDR Testing for Public Health

HIVDR has become a major barrier that hinders ART effectiveness worldwide. A recent meta-analysis and nationally representative surveys have shown that pre-treatment HIVDR to non-nucleoside reverse transcriptase inhibitors (NNRTI) has steadily increased in many LMIC in the last decade, several of which still need to progress to integrase inhibitor-based ART regimens as the preferred first-line option [50,51]. Thus, rising NNRTI HIVDR levels threaten UNAIDS 90-90-90 goals to control the HIV epidemic worldwide [52].

The WHO has proposed an aggressive plan to act upon HIVDR [53], that includes periodic nationally-representative surveillance of pre-treatment and acquired HIVDR using standardized methods [54,55], with the support of a carefully selected and closely monitored laboratory network to guarantee high-quality, reproducible, and comparable HIVDR survey reports worldwide.

The WHO HIVDR Laboratory Network performs genotyping in support of WHO surveys of HIVDR based on a standardized laboratory operational framework [56]. Regional and global analyses depend on the standardization of methods to allow comparison of results from different countries. Assay validation standards have been established and are compulsory for laboratories belonging to the network. Member laboratories participate in an annual Sanger Sequencing EQA proficiency panel testing.

Sanger-based sequencing is the gold standard and most commonly used methodology, both as part of commercial kits or in house-developed and validated protocols; however, NGS methods are being adopted in some laboratories. As mentioned in the previous section in the context of clinical care, specific complexities and standardization issues complicate the reporting of low-abundance resistance variants under the 20% threshold in the public health context. However, potential improvements in the efficiency and cost effectiveness of NGS-based HIVDR genotyping make this approach especially attractive for public health applications, as they often require testing a large number of specimens that could be done in a small number of batches. Thus, while standardization issues are addressed and agreements are reached on low-abundance resistance variant calling and reporting, NGS could be validated as an equivalent method to Sanger (using a 20% variant frequency threshold for reporting). EQA programs already in place for the WHO Global HIVDR Laboratory Network (HIVResNet) can continue assessing NGS-based methods as a Sanger mimic, with no changes needed. The level of rigor of these programs will need to be balanced against feasibility and need issues, especially in the context of LMIC (Table 4). Future requirements, when low-abundance resistance variants under the 20% threshold are considered, will require special considerations on specimen selection, data submission formats and data analysis criteria.

## 5. Challenges in NGS-Based HIVDR Testing Implementation

Even though NGS has great potential to scale up HIVDR testing and lower testing costs, increasing sensitivity for DRM detection, incorporation of NGS technologies for HIVDR testing poses several implementation challenges that significantly hinder and delay their uptake, especially in the context of LMIC. A similar situation has been previously noted in the context of NGS applications for clinical management and surveillance of *Mycobacterium tuberculosis* drug resistance [38]. Many of these challenges are directly associated with general requirements for quality assurance in HIVDR testing, which are strictly necessary to avoid molecular contamination, provide continuous technical support, and ensure continuous operation conditions [56]. Initial economic investments in infrastructure, facility customization, equipment, and personnel training need to be considered as well. Although many of these considerations may seem obvious for experienced laboratories, laboratories with little knowledge on quality assurance processes and molecular biology work, especially in LMIC, could greatly benefit from stating and addressing them before attempting NGS technology implementation.

In some cases, the possibility of outsourcing NGS services to international laboratories may be considered. Nevertheless, LMIC could greatly benefit from implementing NGS techniques locally and technology transfer is desirable. Moreover, in some countries, sample export restrictions exist, as well as budget limitations for sample transport or for subrogation of testing services to core laboratories outside of the country in question. Additionally, it is important for these core laboratories to be included in HIVDR EQA programs.

### 5.1. Infrastructure Requirements

Although a wide range of NGS instruments and protocols exist, some general requirements are necessary for all NGS workflows. At least three separated wet laboratory areas are required to avoid contamination in HIVDR testing, including both Sanger and NGS workflows: a clean space for pre-PCR steps, a sample preparation/DNA extraction area, and a space for amplification/post-amplification procedures. Importantly, specific standardized laboratory procedures must be in place to ensure unidirectional workflow for both personnel and materials and avoid molecular cross-contamination. In addition, adequate spaces for sequencing instruments and designated areas for molecular biology operations and storage of reagents and samples should exist. In connection with this, systems to provide controlled temperature and humidity, as well as continuous power supply, should be in place for the optimal and uninterrupted function of instruments.

Network and internet connections are fundamental for data analysis and result sharing, and often overlooked when considering NGS technologies implementation. Computer hardware and/or other data-storage solutions, data back-ups, and data security measures should also be in place. A minimal laboratory data management system, appropriately suited to laboratory capability and demand for HIVDR tests, is also essential.

### 5.2. Equipment Requirements

The choice of an NGS instrument that fits the needs of the laboratory is essential, for laboratories that aim to conduct in-lab sequencing. Selection should take into account technical requirements such as the preferred sequencing chemistry, error rate, read yield, read length, and time to completion of a sequencing run; but also, practical issues ranging from the price of the instrument, reagents availability and maintenance plans, to the accessibility of technical support, and the existence of local supply distribution chains.

The need for additional equipment should also be considered, including biosafety cabinets for sample handling; PCR hoods for PCR mix preparation; automated DNA extraction instruments, when the volume of sample processing is high; molecular biology grade pipettes (multichannel, automated options are desirable); high precision DNA quantification/analysis instruments such as a fluorometer, a chip-based capillary electrophoresis machine or a real-time PCR instrument; thermocyclers for amplification and incubations; magnetic stands, high-speed microplate shakers, centrifuges, and vortexers for library preparation. Additionally, the availability of −20 °C freezers and 4 °C refrigerators for reagent and sample storage cannot be overlooked.

### 5.3. Logistics and Supply Requirements

For NGS to be cost-effective, high-throughput sample management and pooling for sequencing should be considered. It is noteworthy that the HIVDR testing demand has to be high enough to allow sample multiplexing in a short time in order to provide timely results for patient clinical management with a moderate per-sample cost. This could often be more feasible for reference laboratories providing service for multiple HIV clinics or public health laboratories performing surveillance studies. Ideally, a laboratory information management system (LIMS) or alike administrative process should be in place for monitoring the step-by-step specimen handling procedures to ensure traceability [56]. This includes registration of specimens with unique identifiers, record of the final results in relation to the original specimen identifier, storage of laboratory results, and inventory of remnant specimens.

The existence of local representation, commercial partners, or distributors of companies providing instrument service and reagents is also important for maintaining continuous supply. This implies an understanding of the appropriate international and local regulatory frameworks for importation procedures. Additionally, importers and distributors should have experience in the management and transport of reagents for molecular biology, be able to monitor transport conditions and maintain the cold chain for reagents to perform as expected.

Service support for NGS instruments and technical assistance also needs to be considered. Lack of trained personnel locally could result in long response times for troubleshooting and the need for engineers from other countries to travel. Importantly, the cost of maintenance and service packages should always be considered as part of implementation plans.

Finally, the laboratory should establish standard operating procedures (SOPs) for reagent inventory management, storage, and monitoring shelf-life and quality, foreseeing possible importation delays in order to maintain a continuous supply.

### 5.4. Personnel Requirements

Performing NGS requires skilled personnel in basic molecular biology techniques. In addition, specific training on the NGS workflow and instrument should be provided. Accompaniment and support of companies and commercial partners supplying NGS instruments are often available for the implementation of new applications and continuous personnel training. Additionally, the WHO HIVResNet [56] entails strong potential technical training support for new laboratories, including SOPs transfer, technical support, and personnel training, specifically in HIVDR testing and quality assurance.

### 5.5. Quality Assurance

It is of utmost importance for a HIVDR laboratory to have a quality assurance system in place. Appropriate SOPs should exist, including specimen management, laboratory workflow, technical procedures, quality control, instrument operation and maintenance, result reporting, and biosafety, among others. Many laboratories use in house-developed protocols that need to be validated. Daily processing should be traceable at all points, always including negative and positive controls, with constant monitoring of molecular cross-contamination. The WHO has extensively worked on providing a useful framework for protocol validation and quality control [56].

Participation in EQA programs is highly desirable. However, many issues and questions remain both for the production and distribution of EQA panels and for the establishment of standardized scoring and evaluation systems that address the unique characteristics of NGS. Further discussion on these issues is provided by Lee et al. as part of this special issue [48].

### 5.6. Data and Information Technology Requirements

Providing the appropriate means for data storage, back-up, and processing is a fundamental requirement when planning the implementation of NGS technologies. Considerations of workload, network performance, data security and confidentiality should guide decisions on the best alternatives for data management. Importantly, information technology costs should also be contemplated within implementation plans.

The availability of free public web-based solutions for HIVDR analysis of NGS data [10] that do not require personnel with bioinformatics skills is highly advantageous and a remarkable achievement in the field. Even though no software licensing is required, a reliable and fast internet connection would be a requirement for data analysis using these platforms.

## 6. Conclusions

NGS is a powerful tool for HIVDR assessment with higher sensitivity, higher efficiency, and lower costs when batched specimens are being processed, as compared to Sanger sequencing. Standardization of NGS poses an important challenge, considering the many potential sources of variations and possible bias in the NGS laboratory workflow, i.e., starting material and sample type, PCR amplification requirements, library preparation method, instrument and sequencing chemistry-inherent error, and data analysis options and limitations. Implementation of NGS for HIVDR genotyping in LMIC may be especially challenging due to infrastructure and equipment requirements and costs, laboratory and clinical personnel training, logistics and supply chains, service availability, and quality assurance. The level of rigor required for clinical NGS-based HIVDR testing may need to be more flexible in LMIC, considering implementation limitations. The establishment of EQA programs may be key to achieve homogeneity and reliability in NGS-based HIVDR genotyping and leverage on its advantages.

## Figures and Tables

**Figure 1 viruses-12-00617-f001:**
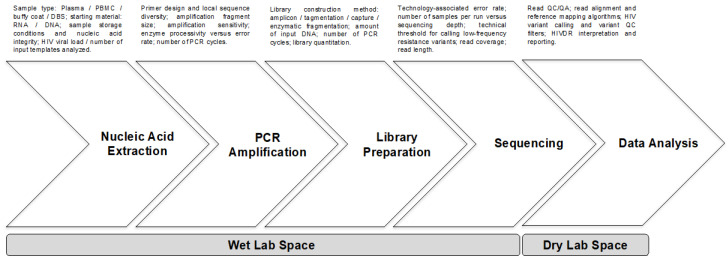
General steps of the NGS workflow for HIVDR genotyping. Common steps for most of the second-generation NGS workflows are shown, including the main sources of variation and possible bias associated with each step. NGS—next generation sequencing; PBMC—peripheral blood mononuclear cells; DBS—dried blood spots; PCR—polymerase chain reaction; QC—quality control; QA—quality assurance; HIVDR—HIV drug resistance.

**Table 1 viruses-12-00617-t001:** Comparison between commercially available NGS-based solutions for HIVDR genotyping and in house-developed protocols.

Test (Manufacturer)	HIV Gene Coverage	NGS Platform	Analysis Software	Reported Sensitivity	Regulatory Status	Cost Per Sample
DeepChek-HIV (ABL, Luxemburg)	PR, RT, IN	Ion Torrent PGM, Illumina MiSeq	ViroScore-HIV/DeepChek-HIV	1%	CE-IVD for software, RUO for kits	$120
DeepGen (CWRU, USA)	PR, RT, IN	Ion Torrent PGM, Illumina MiSeq	DeepGen	1%–5%	Core service ^a^	$100
Sentosa SQ HIV (Vela Diagnostics, Singapore)	PR, RT, IN	Ion Torrent PGM	Sentosa SQ	10%	TGA CE-IVD HAS Class C IVD FDA Class II IVD	$400
In-house assays	PR, RT, IN	Ion Torrent PGM, Illumina MiSeq	HyDRA, PASEQ, MiCall, HIVmmer	Variable	-	$50–150

^a^ Validated test to be used in a CAP/CLIA-certified laboratories [13,14]. Abbreviations: ABL: Advanced biological Laboratories; CWRU: Case Western Reserve University; PR: protease; RT: Reverse transcriptase; IN: Integrase; PGM: Personal Genome Machine; CE-IVD: European CE Marking for In Vitro Diagnostic (IVD) devices; RUO: Research use only; TGA: Therapeutic Goods Administration in Australia; CAP/CLIA: College of American Pathologists/Clinical Laboratory Improvement Amendments; FDA: US Food and Drug Administration; HSA: Singapore Health Science Authority.

**Table 2 viruses-12-00617-t002:** Examples of commercially available kits for NGS library preparation.

Library Preparation Kit	Manufacturer	System Compatibility	Principle	Minimum DNA Input Quantity
Nextera XT	Illumina	Illumina	Tagmentation	1 ng
Nextera DNA Flex	Illumina	Illumina	Tagmentation	1 ng
Ion Xpress Plus Fragment	ThermoFisher	Ion Torrent	Enzymatic fragmentation	100 ng
MuSeek	ThermoFisher	Ion Torrent, Illumina	Tagmentation	50 ng
NEXTFLEX DNA Seq	PerkinElmer	Ion Torrent, Illumina	Enzymatic fragmentation	1 ng
KAPA HyperPlus	Roche	Illumina	Enzymatic fragmentation	50 ng
NEBNext Ultra	New England BioLabs	Ion Torrent, Illumina	Enzymatic fragmentation	100 pg

**Table 3 viruses-12-00617-t003:** General features of the two currently most commonly used NGS platforms in HIVDR genotyping.

Instrument (Manufacturer)	Chemistry	Detection	Data Output	Maximum Read Length	Reported Accuracy ^a^/Error Rate	Sequencing Time	Instrument Cost (USD)	Strengths	Weaknesses
MiSeq (Illumina)	Sequencing by synthesis (bridge PCR)	Fluorescence	0.3–15 Gb; 2–50 million reads	2 × 300 bp	Mostly > Q30/0.8%	4–55 h	128,000	Accuracy, read length	Long run time
PGM (ThermoFisher)	Sequencing by synthesis (emulsion PCR)	Semi-conductor	0.03–2 Gb; 0.4–5.5 million reads	400 bp	Mostly > Q20/1.7%	2–10 h	80,000	Short run time, read length	Low throughput, homopolymers

^a^ A base with Q30 (Phred-like Q) score has a probability of 1 in 1000 and a base with Q20 a probability of 1 in 100 of an incorrect base-call. Modified from [36,38].

**Table 4 viruses-12-00617-t004:** NGS implementation in resource-poor settings, feasibility, and challenges.

Challenges	Solutions
Cost	Generate economies of scale: high-throughput sample processing;
Instrument access	Use of core facilities; negotiations with suppliers.
Comparability	Using Sanger mimic conditions with conservative thresholds (20%) to report DRMs; Recommendations of the First “Winnipeg Consensus” [10].
Bioinformatics and data analysis	Specialized, freely available, or low-cost pipelines.
Personnel training/retention	Support of laboratories within WHO HIVResNet. Support from instrument manufacturers and suppliers.
Quality assurance	Support from WHO HIVResNet; Search for additional support from other leading international agencies, such as the Public Health Agency of Canada or the US Centers for Disease Control and Prevention.

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
