# Peer review of "Next-Generation Sequencing for HIV Drug Resistance Testing: Laboratory, Clinical, and Implementation Considerations"

_viruses, 2020, doi:10.3390/v12060617_

Round 1

Reviewer 1 Report

This is a paper about the existing challenges to use NGS for determining HIV drug resistance for both clinical care and public health purposes. The main conclusion was that NGS is a powerful tool for HIVDR assessment with higher sensitivity, higher efficiency and lower costs when batched specimens are being processed, as compared to Sanger sequencing. However, there are potential sources of variations and possible bias in the NGS laboratory workflow that should be considered being the standardization of NGS that represents an important challenge for the future times.

This paper addresses several important issues, all being very explicit and very comprehensive in general, however from my point of view, it’s just a review that doesn’t bring anything new about this subject, since already exist several published reviews on this subject.

There are some revision/ amendments that must be made:

Line 103- figure 1 is not well formatted and it has several acronyms that are not explained

Line 211- this paragraph is not very explicit, it may bring doubts to the reader and can generate several interpretations. I propose that should be rewritten.

Line 214- An interesting hypothesis to assess the threshold to determine less abundant populations would be to analyze the sequencing of the same patient before the introduction of the therapy and weeks after using it in order to understand the impact of minority populations. I suggest because I think it is important to have some reference to how to do it.

Line 344- It is worth mentioning that you can pay for facilities in other countries like China, that is, you do not need local service. I think it is a possibility that must be considered.

Line 380-The author assumes the need for a wet-lab, which is not entirely true, since you can pay for facilities and only have the raw data. Sometimes the cost benefit is higher. It would be worthwhile to bring both questions to the debate.

Author Response

This paper addresses several important issues, all being very explicit and very comprehensive in general, however from my point of view, it’s just a review that doesn’t bring anything new about this subject, since already exist several published reviews on this subject.

A: We appreciate the Reviewer’s point of view and understand that some of the issues raised in this review may seem obvious or not novel for some people working in experienced laboratories, especially in developed countries. However, we do believe that these points need to be underscored for developing countries who are planning to implement NGS technologies locally, as some issues can be overlooked. We also believe that many of the methodological, sample-related and analysis-associated issues raised in the manuscript are current problems being faced by experts in the field, and are of interest to more advanced groups and laboratories also. We have included this comment in the introduction (lines 75-81). Also, a comment regarding the Reviewer’s point had been included in the original version in line 339-342 (357-360 in the revised version): “Although many of these considerations may seem obvious for experienced laboratories, laboratories with little knowledge on QA processes and molecular biology work, especially in LMIC, could greatly benefit from stating and addressing them before attempting NGS technology implementation”.

There are some revision/ amendments that must be made:

Line 103- figure 1 is not well formatted and it has several acronyms that are not explained

A: Formatting of Figure 1 was lost when transforming to pdf. We have uploaded an updated version of the figure and have included the definition of all acronyms in the figure legend.

Line 211- this paragraph is not very explicit, it may bring doubts to the reader and can generate several interpretations. I propose that should be rewritten.

A: This sentence is trying to call the reader’s attention to the fact that technical thresholds to call low-abundance variants are different to clinical thresholds to define whether a low-abundance HIVDR variant is relevant for ART outcome. Even if the accuracy to call low-abundance variants is high at very low thresholds, these variants may not be of clinical significance. We have added this sentence to clarify (lines 227-229 of the revised version).

Line 214- An interesting hypothesis to assess the threshold to determine less abundant populations would be to analyze the sequencing of the same patient before the introduction of the therapy and weeks after using it in order to understand the impact of minority populations. I suggest because I think it is important to have some reference to how to do it.

A: We thank the Reviewer for this comment, but we believe this paragraph refers to a different concept. Here, we are not describing how to assess the clinical relevance of low-abundance drug resistance variants. The strategy described is a proposal to select an appropriate technical threshold to call low abundance variants, on a sample per sample basis, using the proportion of unusual mutations observed in each sample. We hope this answer is satisfactory and respectfully ask to leave this concept as it is.

Line 344- It is worth mentioning that you can pay for facilities in other countries like China, that is, you do not need local service. I think it is a possibility that must be considered.

A: We appreciate the Reviewer’s concern and have included a comment on the possibility of outsourcing NGS services in section 5 (lines 361-366). Nevertheless, an important point raised by this review is that developing nations could greatly benefit from implementing NGS techniques locally and technology transfer is important. In many occasions, sample export restrictions exist, as well as budget restrictions for covering sample transport or for subrogation of HIVDR testing to external core laboratories, which are not in the country in question. We have added this comment to the text.

Line 380-The author assumes the need for a wet-lab, which is not entirely true, since you can pay for facilities and only have the raw data. Sometimes the cost benefit is higher. It would be worthwhile to bring both questions to the debate.

A: We agree with the Reviewer, though many developing countries have expressed their intention to implement NGS technologies for HIVDR testing in situ. This requirement often comes in conjunction with other testing necessities that could leverage on implementation of this technology, enriching the technical capacity of the countries. Although the possibility of outsourcing the service should be considered, it is also true that technology transfer and local development is desirable. To address this comment, we added this concept in the text (lines 361-366).

Reviewer 2 Report

This well-written review summaries the important considerations for introducing and comparing Sanger sequencing protocols with common NGS platforms.

This review may be of assistance for laboratories, especially in low and middle income settings, which consider the introduction of HIV resistance testing and need advice for the appropriate system choice.

The review covers well all steps from sample preparation, including sample properties (viral load and type) to laboratory processes and up to the computer-based analysis and reporting (including QC and standardisation).

The manuscript covers most aspects, but the authors may want to bring to the attention of the reader some additional crucial issues directly arising from low viremia samples, such as:

  • the inherently low genetic diversity of samples with a low copy number (here, a very low cutoff can even suggest a false diversity)
  • Also, a statement on the currently missing clinical meaningfulness of low percentage mutants should be added: Especially for LMIS it appears important to mention that the "time to failure" has not at all been defined for mutants that are found at low frequency. Such a statement will help to avoid too early switching as soon as a potentially deleterious mutation has been detected by testing = clinical context must be considered.

minor comments:

in Figure 1 the text (in arrows) has been cut

in table 1: the column "cost" does not define if these are "cost per test",  "cost per run" etc.

Table 3: format of columns is odd: text line breaks within words - to be corrected

Author Response

The manuscript covers most aspects, but the authors may want to bring to the attention of the reader some additional crucial issues directly arising from low viremia samples, such as:

  • the inherently low genetic diversity of samples with a low copy number (here, a very low cutoff can even suggest a false diversity)

A: We agree with the Reviewer that this is an important point and, as suggested, have added a comment in section 2.1 (lines 152-155 of the revised version) regarding samples with low copy number.

  • Also, a statement on the currently missing clinical meaningfulness of low percentage mutants should be added: Especially for LMIS it appears important to mention that the "time to failure" has not at all been defined for mutants that are found at low frequency. Such a statement will help to avoid too early switching as soon as a potentially deleterious mutation has been detected by testing = clinical context must be considered.

A: We thank the Reviewer for this important point. In lines 55-59 (58-62 in the revised version), we had already mentioned about the lack of a clinically relevant threshold for low-abundance variants. We then discussed this further in lines 211-213 (225-227 in the revised version) and 256-263 (277-283 in the revised version). We have added an additional comment in section 3 (lines 274-277), as suggested by the Reviewer.

minor comments:

in Figure 1 the text (in arrows) has been cut

A: The format of Figure 1 was lost when transforming to pdf. We have uploaded an updated version of the figure.

in table 1: the column "cost" does not define if these are "cost per test", "cost per run" etc.

A: We have corrected the column title as: “Cost per sample”

Table 3: format of columns is odd: text line breaks within words - to be corrected

A: Format of Table 3 was lost when transforming to pdf. We have reformatted the table.

Round 2

Reviewer 1 Report

Thank you very much for the adding revisions.